# Survival Rate and Prosthetic and Sinus Complications of Zygomatic Dental Implants for the Rehabilitation of the Atrophic Edentulous Maxilla: A Systematic Review and Meta-Analysis

**DOI:** 10.3390/biology10070601

**Published:** 2021-06-29

**Authors:** David Gutiérrez Muñoz, Caterina Obrador Aldover, Álvaro Zubizarreta-Macho, Héctor González Menéndez, Juan Lorrio Castro, David Peñarrocha-Oltra, José María Montiel-Company, Sofía Hernández Montero

**Affiliations:** 1Department of Implant Surgery, Faculty of Health Sciences, Alfonso X el Sabio University, 28691 Madrid, Spain; dgutierr@uax.es (D.G.M.); mobraadr@myuax.com (C.O.A.); hgonzmen@uax.es (H.G.M.); jlorrio@uax.es (J.L.C.); shernmon@uax.es (S.H.M.); 2Department of Surgery, Faculty of Medicine and Dentistry, University of Salamanca, 37008 Salamanca, Spain; 3Department of Stomatology, Faculty of Medicine and Dentistry, University of Valencia, 46010 Valencia, Spain; david.penarrocha@uv.es (D.P.-O.); jose.maria.montiel@uv.es (J.M.M.-C.)

**Keywords:** zygomatic implants, prosthetic rehabilitation, maxillary sinus, sinusitis, implant failure, survival

## Abstract

**Simple Summary:**

Zygomatic dental implants have been proposed as an alternative to atrophic total edentulous maxillae rehabilitation with the necessity of bone grafting procedures. However, surgical, prosthetic, and maxillary sinus complications have been associated with this surgical procedure. Therefore, it is necessary to produce a systematic review and meta-analysis that provides evidence associated with the prognosis when using zygomatic dental implants as an alternative to atrophic total edentulous maxillae rehabilitation.

**Abstract:**

The aim of this systematic review and meta-analysis was to analyze and compare the survival rate and prosthetic and sinus complications of zygomatic dental implants for the rehabilitation of the atrophic edentulous maxilla. Materials and methods: We conducted a systematic literature review and meta-analysis, based on the Preferred Reporting Items for Systematic Reviews and Meta-Analyses (PRISMA) recommendations, of clinical studies that evaluated the survival rate and prosthetic and sinus complications of zygomatic dental implants for the rehabilitation of the atrophic edentulous maxilla. Four databases were consulted during the literature search: Pubmed–Medline, Scopus, Embase, and Web of Science. After eliminating duplicate articles and applying the inclusion criteria, 46 articles were selected for the qualitative analysis and 32 for the quantitative analysis. Results: Four randomized controlled trials, 19 prospective clinical studies, 20 retrospective studies, and 3 case series were included in the meta-analysis. Conventional dental implants failure (*n* = 3549) were seen in 2.89% (IC-95% 1.83–3.96%), while zygomatic dental implants failure (*n* = 1895) were seen in 0.69% (IC-95% 0.21–1.16%). The measure of the effect size used was the Odds Ratio, which was estimated at 2.05 with a confidence interval of 95% between 1.22 and 3.44 (z test = 2.73; *p*-value = 0.006). The failure risk of conventional dental implants is 2.1 times higher than that of zygomatic dental implants. Slight heterogeneity was determined in the meta-analysis between 23 combined studies (Q test = 32.4; *p*-value = 0.070; I^2^ = 32.1%). Prosthetic complications were recorded in 4.9% (IC-95% 2.7–7.3%) and mild heterogeneity was observed in a meta-analysis of 28 combined studies (Q test = 88.2; *p*-value = 0.001; I^2^ = 69.4%). Sinus complications were seen in 4.7% (IC-95% 2.8–6.5%) and mild heterogeneity was observed in a meta-analysis of 32 combined studies (Q test = 75.3; *p*-value = 0.001; I^2^ = 58.8%). Conclusions: The high survival rate and low prosthetic and sinus complications related to zygomatic dental implants suggest the use of zygomatic dental implants for the rehabilitation of the atrophic edentulous maxilla.

## 1. Introduction

The rehabilitation of extremely atrophic, fully edentulous maxillae is a concern and constitutes a challenge for dental professionals due to the lack of bone availability, which influences the placement of conventional length dental implants [1]. Various therapeutic alternatives have been proposed to rehabilitate the atrophic maxilla by bone-augmentation procedures to increase the bone availability, allowing implant-supported rehabilitation, including grafting procedures, sinus lift, and apposition graft with or without Le Fort I osteotomy, with success rates of 60–90% [2,3,4]. However, most of these approaches require delayed approaches and two-stage procedures, including bone grafts that increase the risk of potential postoperative complications [5]. In addition, a higher implant failure rate has been associated with a lack of bone availability and/or low, inadequate bone density in edentulous patients with atrophic maxilla [6,7]. Moreover, bacterial infection has been correlated to the development of peri-implant disease; thus, it is important to analyze the bacterial biotypes and biomarkers associated with implant failure. Isola et al. reported significantly higher serum and salivary Galectin-3 levels in patients affected with periodontitis compared with healthy subjects. They also reported that periodontitis and Endothelin-1 were significant predictors of serum and salivary Galectin-3 levels, respectively [8]. Furthermore, Ghassib et al. conducted a systematic review and meta-analysis and reported that pro-inflammatory cytokines in peri-implant crevicular fluid, such as interleukin-1β and interleukin-6, can be used as adjunct tools to clinical parameters to differentiate healthy patients from peri-implant mucositis and peri-implantitis [9]. Therefore, zygomatic dental implants have been proposed as an alternative to atrophic fully edentulous maxillae rehabilitation with the necessity of bone-grafting procedures [10]. The zygomatic implants approach has been used in conjunction with conventional-length dental implants in patients with severe resorption of the maxilla, with a survival rate of 96–100% [11,12,13]. Unfortunately, postoperative complications have been reported in terms of the effect on the maxillary sinus, especially when placing intrasinusal zygomatic dental implants. A sinusitis incidence rate of 5–6% has been reported (range: 0–26.6%); however, antibiotic therapy has been shown to be broadly effective in all patients [14,15]. Prosthetic complications have also been reported in implant-supported restorations using zygomatic dental implants. Prosthetic complications have also been reported, related to adjustments of the retention elements of overdentures, the fracture of fixed dental prostheses, mucosal overgrowth, or hyperplasia and discomfort [16].

The aim of this systematic review and meta-analysis was to analyze and compare the survival rate and prosthetic and sinus complications of zygomatic dental implants for the rehabilitation of the atrophic edentulous maxilla, with a null hypothesis (H_0_) stating that there would be no difference between the survival rate and prosthetic and sinus complications of zygomatic dental implants and conventional-length dental implants for the rehabilitation of the atrophic edentulous maxilla.

## 2. Materials and Methods

### 2.1. Study Design

A bibliographic search was conducted following the PRISMA (Preferred Reporting Items for Systemic Reviews and Meta-Analyses http://www.prisma-statement.org, accessed on 17 June 2021) guidelines for systematic reviews and meta-analyses (PROSPERO registration number: CRD42021226821). The review also fulfilled the PRISMA 2009 Checklist [17].

### 2.2. Focused Question

The PICO (population, intervention, comparison, outcome) question was “What is the survival rate, and what are the prosthetic and sinus complications, of zygomatic dental implants for the rehabilitation of the atrophic edentulous maxilla?” with the following components: population: atrophic edentulous maxilla patients treated with zygomatic dental implants; intervention: rehabilitation of the atrophic edentulous maxilla through zygomatic dental implants; comparison: zygomatic dental implants and conventional dental implants; and outcomes: survival rate and prosthetic and sinus complications.

### 2.3. Databases and Search Strategy

An electronic search was conducted in the following databases and gray literature: PubMed; Scopus; Embase, Web of Sciences and OpenGrey (www.opengrey.eu, accessed on 17 June 2021) (A.Z.-M J.M.M.C). The search covered all the literature published internationally up to June 2020. The search included seven medical subject heading (MeSH) terms: “zygomatic implants”, “survival rate”, “prognosis”, “implant failure”, “prosthetic rehabilitation”, “complications”, “maxillary sinus”, and “sinusitis.” The Boolean operators applied were OR and The search terms were structured as follows: ((“zygomatic implants”) AND (“survival rate”) OR (“prognosis”) OR (“implant failure”) AND (“prosthetic rehabilitation”) AND (“complication”) AND (“maxillary sinus”) OR (“sinusitis”)). Two researchers (S.H.M. and A.Z.-M.) conducted the database searches in duplicate, independently. Titles and abstracts were selected by applying the inclusion and exclusion criteria.

### 2.4. Study Selection

Titles and abstracts were selected after applying inclusion and exclusion criteria by two authors (C.O.A. and J.R.G.R.).

Inclusion criteria: studies recorded in databases as prospective randomized clinical trials (RCTs), retrospective studies, and case series from five patients. The review was not restricted to RCTs because of the paucity of studies with this experimental design and external validity, but also to provide a complete picture of the topic.

Samples of patients aged 18 years old or over; patients treated with zygomatic dental implants to rehabilitate atrophic edentulous maxilla; follow-up period of at least 3 months. No restriction was placed on the year of publication or language.

Exclusion criteria: systematic literature reviews, clinical cases, case series of up to five patients, and editorials; studies including patients under the age of 18; studies with samples of five or fewer patients. The following data were extracted from each article by two authors (C.O.A. and J.R.G.R.): author and year of publication; title and journal in which the article was published; sample size (*n*); follow-up time and success rate, periapical healing reduction, and bone density. Studies that analyzed implant failure rate and prosthetic and sinus complications were included in the systematic review and network meta-analysis.

### 2.5. Data Extraction and Study Outcomes

Data extraction was conducted in duplicate (by C.O.A. and J.R.G.R.) using predefined Excel spreadsheets and accounting for the following items: author and year, study type, sample size, follow-up in months, implant failures, prosthetic complications, and presence of sinusitis.

### 2.6. Methodological Quality Assessment

The risk of bias in the studies selected for review was assessed by two authors (M.P.D; D.P.O) using the Jadad scale for methodological quality assessment of clinical trials. The Jadad scale consists of five items that evaluate randomization, researcher and patient blinding, and description of losses during follow-up producing a score of 0–5; scores of less than three are considered to indicate low quality [18]. The level of agreement between evaluators was determined using Kappa scores.

### 2.7. Quantitative Synthesis—Meta-Analysis

The statistical data collection and analysis were conducted by two authors (A.Z.-M. and J.M.M.-C.). The studies included for the meta-analysis were combined using a random-effects model with various methods according to the estimated effect size. The inverse method of variance was used to estimate the root apex location success rate, the Mantel Haenszel method for the Odds Ratio (OR), and the inverse method of variance for the mean difference. For all the estimated variables, a 95% confidence interval was calculated. Heterogeneity between the combined studies was assessed using the Q test (*p*-value < 0.05) and quantified with the I^2^, with a slight heterogeneity if it is 25–50%, moderate at 50–75%, and high if >75%. Statistical significance was assessed using the Z test (*p*-value < 0.05). Meta-analyses were represented with a forest plot. Publication bias was assessed using the Trim and Fill adjustment method, represented with Funnel plots. The R software was employed for meta-evidence analysis.

## 3. Results

### 3.1. Flow Diagram

The initial electronic search identified 37 articles in PubMed, 40 in Web of Sciences, 31 in Embase, 21 in Scopus, and none in the gray literature. Of the 129 works, 32 were discarded as duplicates. After reading the titles and abstracts, a further 35 were eliminated, leaving a total of 62. A further 15 were rejected as they failed to fulfill the following inclusion criteria: they did not include survival rate data, did not include prosthetic or sinus complications data, or presented a minimum follow-up time of 3 months. A final total of 46 articles were included in the qualitative synthesis. Thirty-two articles were included in the quantitative synthesis, as these included all the data and variables required (Figure 1).

### 3.2. Qualitative Analysis

Of the 46 articles included, 4 were randomized clinical trials [19,20,21,22], 19 were nonrandomized clinical trials [10,11,14,23,24,25,26,27,28,29,30,31,32,33,34,35,36,37,38], 20 were retrospective studies [7,8,13,39,40,41,42,43,44,45,46,47,48,49,50,51,52,53,54], and 3 were case series [55,56,57]. In addition, 31 articles compared the success rate of conventional and zygomatic dental implants [7,8,9,11,13,17,19,21,22,23,24,25,26,27,28,29,32,33,34,36,38,41,42,43,44,46,48,53,54,55]. Fifteen articles described an intrasinusal placement of the zygomatic dental implants [1,7,13,20,21,24,27,29,32,33,40,42,47,54], six articles described an extrasinusal placement of the zygomatic dental implants [11,23,34,36,43,46], five articles described the sinus slot placement technique of the zygomatic dental implants [45,48,50,52,55], three articles described intra and extrasinusal placement of the zygomatic dental implants [26,35,39], one article described the sinus slot technique and intrasinusal placement of the zygomatic dental implants [9], and two articles did not describe the placement technique of the zygomatic dental implants [8,37]. Twenty-eight articles analyzed the prosthetic complications [7,8,10,11,14,17,18,19,21,22,23,24,25,26,31,34,36,38,41,42,43,44,46,48,49,50,54] and thirty-two articles described the sinus complications [7,11,13,20,21,23,24,25,26,27,29,32,33,34,35,36,37,38,39,40,41,42,43,45,46,47,48,50,52,54,55]. Most of the studies presented a follow-up time of approximately 36 months, ranging from 3 months in the study by Fernández et al., 2015 [20], to 163 months in Agbara’s study from 2017 [37]. The results are presented in Table 1.

### 3.3. Quality Assessment

The results of the methodological quality assessment using the Jadad scale were performed by one author (A.Z.-M.) and are shown in Table 2. The Jadad scale returned 23 articles as “not applicable”, because 20 were retrospective [7,8,13,37,38,39,40,41,42,43,44,45,46,47,48,49,50,51,52] and 3 were case series [53,54,55], and the authors of these articles did not blind or randomize the studies. Two articles [17,20] obtained scores of five, indicating high methodological quality. Again, quality was most frequently compromised by a failure to fulfill items related to the subject, treatment, or measurement blinding.

### 3.4. Quantitative Analysis

#### 3.4.1. Failure Rate of Zygomatic and Conventional Dental Implants

The incidence of implant failure of conventional dental implants (*n* = 3549) has been estimated at 2.89% (CI-95% 1.83–3.96%), while the incidence of implant failure of zygomatic dental implants (*n* = 1895) has been estimated at 0.69% (CI-95% 0.21–1.16%). The follow-up time of the studies selected was 3–163 months.

Thirty-one studies [7,8,9,11,13,17,19,21,22,23,24,25,26,27,28,29,32,33,34,36,38,41,42,43,44,46,48,53,54,55] that compared the incidence of dental implant failure between conventional and zygomatic dental implants were included in the meta-analysis and combined using a random effects model with the Mantel–Haenszel method. The effect size measure used was the Odds Ratio, which was estimated at 1.33 with a 95% confidence interval between 0.79 and 2.23 (z-test = 1.09; *p*-value = 0.278). The risk of implant failure is 1.3 times greater with conventional than with zygomatic dental implants. Meta-analysis has shown a slight heterogeneity between the combined studies (Q test = 44.48 *p*-value = 0.039; I^2^ = 33.1%) (Figure 2).

#### 3.4.2. Incidence of Prosthetic Complications in Patients with Zygomatic Implants

Twenty-eight studies [7,8,10,11,14,17,18,19,21,22,23,24,25,26,31,34,36,38,41,42,43,44,46,48,49,50,54] with a total of 921 patients were combined using a random effects model (inverse variance method), estimating an incidence of prosthetic complications of 4.9% with a 95% confidence interval between 2.7% and 7.3% of patients with zygomatic implants. The meta-analysis detected moderate heterogeneity between the combined studies (Q-test = 88.2; *p*-value = 0.0001; I^2^ = 69.4%) (Figure 3).

#### 3.4.3. Incidence of Sinusitis in Patients with Zygomatic Implants

Thirty-two studies [7,11,13,20,21,23,24,25,26,27,29,32,33,34,35,36,37,38,39,40,41,42,43,45,46,47,48,50,52,54,55] with a total of 1119 patients, combined using a random effects model (inverse variance method), obtained an estimate of the incidence of sinusitis of 4.7% with a 95% confidence interval between 2.8% and 6.5% of patients with zygomatic implants. The meta-analysis detected moderate heterogeneity between the combined studies (Q test = 75.3; *p*-value < 0.0001; I^2^ = 58.8%) (Figure 4).

The cumulative incidence of sinus complications in patients with zygomatic implants placed using an intrasinusal technique was 7.2% (CI-95% 4.6–9.8%), significantly higher (Q test between groups = 8.85; *p*-value = 0.0029) than with the extrasinusal technique, which showed a cumulative incidence of 1.8% (CI-95% 0.0–4.2%) (Figure 5).

### 3.5. Publication Bias

No study has been added to the 32 studies initially combined, using the Trim and Fill method to obtain symmetry in the funnel plot. The Odds Ratio estimation of dental implant failure, adjusted by the Mantel–Haenszel random-effects model, was 1.33 (95% CI between 0.79 and 2.23), showing no difference from the initial Odds Ratio estimation. Figure 6 shows the two funnel plots (initial and adjusted). These data indicate the absence of publication bias.

## 4. Discussion

The results obtained in the present study led us to reject the null hypothesis (H_0_), stating that there would be no difference between the survival rate and prosthetic and sinus complications of zygomatic dental implants for the rehabilitation of the atrophic edentulous maxilla.

The study showed that zygomatic dental implants had a lower failure rate than conventional-length dental implants. In addition, the prosthetic rehabilitations of zygomatic dental implants showed low prevalence values. Finally, the maxillary sinus complications of the atrophic edentulous maxilla rehabilitated by zygomatic dental implants also showed low prevalence values.

The meta-analysis showed a predictable outcome for the zygomatic dental implants, with a failure rate of 0.69% (CI-95% 0.21–1.16%) at 4–120 months follow-up. Some studies have reported the influence of the dental implant length on the long-term outcome of dental implants; therefore, the survival rate of zygomatic dental implants is higher than that of conventional-length dental implants (2.89% (CI-95% 1.83–3.96%)). A higher length of zygomatic dental implants creates a larger osseointegration surface that promotes the integration and stability of the zygomatic dental implants and improves the distribution of occlusal loads, since the length of zygomatic dental implants ranges from 30 mm to 52.5 mm [24] and conventional-length dental implants’ length ranges from 10 mm to 15 mm [38,42]. Some authors have reported the combined use of zygomatic dental implants in the atrophic edentulous posterior maxilla and conventional length dental implants in the anterior area [17,18,20,21,22,24,27,29,32,35,37,38,43,45,48,50]. Bedrossian et al. (2010) proposed treatment guidelines based on the bone availability and recommended placing four conventional-length dental implants in zones I and II; conventional-length dental implants in zones I, II, and III; combined conventional-length dental implants and zygomatic dental implants in zone I only; and four zygomatic dental implants in cases of insufficient bone availability [27]. The location of conventional-length dental implants and their shorter length simplify the explantation procedure and the posterior bone regeneration technique if necessary. Additionally, bacterial contamination has been highlighted as a relevant factor related to periodontal disease and implant failure; therefore, advances in microbial molecular diagnostics have allowed for better identification and thus a greater understanding of the causative agents and related biomarkers involved in both diseases [8,9]. In addition, Bedrossian et al. described the ideal number and location of zygomatic dental implants and conventional-length dental implants for atrophic edentulous maxilla rehabilitation by placing a minimum of two premaxillary conventional-length dental implants in the canine position, or ideally four premaxillary conventional-length dental implants in the canine and the central incisor positions and two zygomatic dental implants introduced into the second premolar area [30].

Zygomatic dental implants’ placement is still a challenge that poses risks because the lack of bone availability in the atrophic edentulous maxilla requires longer implants to attach to distant anatomical structures and can lead to clinical complications [35]. Therefore, both static and dynamic navigation systems have been widely used in dental implants [58,59,60,61,62,63,64,65,66,67]. Computer-aided static navigation systems have shown a mean horizontal deviation at the coronal entry point and apical endpoint of 1.2 mm (1.04–1.44 mm) and 1.4 mm (1.28–1.58 mm), respectively, and a mean angular deviation of 3.5° (3.0–3.96°). However, computer-aided dynamic navigation systems have demonstrated lower deviation values at the coronal entry point (0.71 ± 0.40 mm), apical endpoint (1.00 ± 0.49 mm), and angular deviation (2.26 ± 1.62°). Therefore, these results have encouraged us to apply computer-aided navigation techniques to zygomatic dental implants in order to improve the accuracy of zygomatic dental implants and prevent intraoperative complications [33,36].

Prosthetic rehabilitations of the zygomatic dental implants have demonstrated a low prosthetic complication incidence (4.9% (CI-95% 2.7–7.3%)), regardless of the prosthetic treatment. Many prosthetic treatments have been proposed to rehabilitate zygomatic dental implants, but problems have been reported, such as allergy to the gold alloy of the overdenture bar [22], losing the zygomatic implant gold screw, fracture of the gold screw [24], fracture of the metal–porcelain prostheses, fracture of the abutment screw [26], fracture of the framework, losing the gold zygomatic dental implant screws, fracture of the gold screws, losing the abutment screw, fracture of ceramic prosthetic teeth, fracture of the resin prostheses, disconnected abutments [1], partial fractures in the denture around the zygomatic implant cylinder [8], fractures or detachments of one or more acrylic teeth, fracture of provisional prosthesis [36], and fracture of the metal bar [34].

Most authors reported an absence of sinus pathology related to zygomatic dental implants; however, some of them were placed in an extrasinusal location. The authors also reported that the sinusitis observed after the placement of zygomatic dental implants was resolved favorably after the administration of antiseptics (chlorhexidine 0.2%), antibiotics (amoxicillin and clavulanic acid 1000 mg), and corticosteroids. The results obtained in this meta-analysis showed a low sinus complication incidence (4.7% (CI-95% 2.8–6.5%)) regarding the relationship between the zygomatic dental implants and the maxillary sinus.

The results of the present work can be extrapolated to those patients with atrophied maxilla requiring full-arch rehabilitation by means of zygomatic implants. There is a need for a larger body of evidence with more randomized studies, until today scarce in the literature. Thus, more studies are warranted; of special interest are those implementing new technologies (e.g., CAD-CAM, intraoral scanners, guided surgery) or concomitant regenerative procedures.

In addition, the studies selected in the present systematic review and meta-analysis showed low methodological quality; therefore, the authors highlight the necessity of improving the methodological design for future studies. Moreover, the present review has endeavored to summarize the best available evidence, but not always the least biased. The majority of the articles showed a risk of bias, which is inherent to the observational design. Additionally, blinding methods were frequently not applied, which increased the risk of bias. Despite the abovementioned drawbacks, the inconsistency of the results proves to be low to moderate, with I^2^ values < 75%. Moreover, the hints of meta-bias were properly inspected using funnel plots and showed a symmetrical distribution. All this, together with the comprehensive electronic searches and prospective protocol registration, increases our confidence in the review findings.

## 5. Conclusions

Meta-evidence suggests that zygomatic implants have higher survival rates than conventional implants in patients with severely atrophied maxilla; however, zygomatic implants are not recommended as a first treatment option. The incidence of prosthetic complications and sinusitis is low. The impact of covariates such as surgical technique on biological complications requires further study.

## Figures and Tables

**Figure 1 biology-10-00601-f001:**
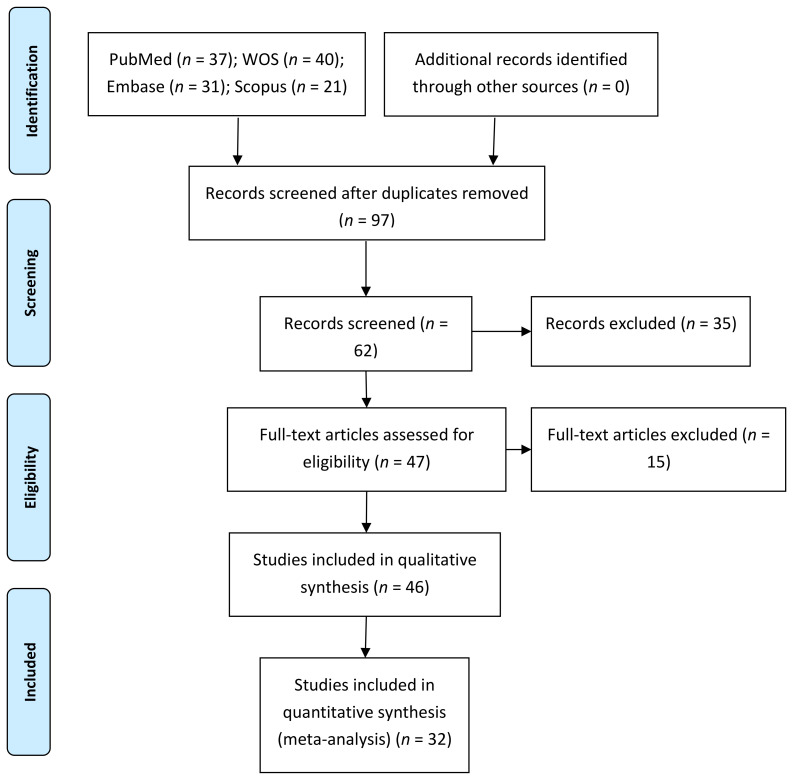
Preferred reporting items for systematic reviews and meta-analyses (PRISMA) flow diagram.

**Figure 2 biology-10-00601-f002:**
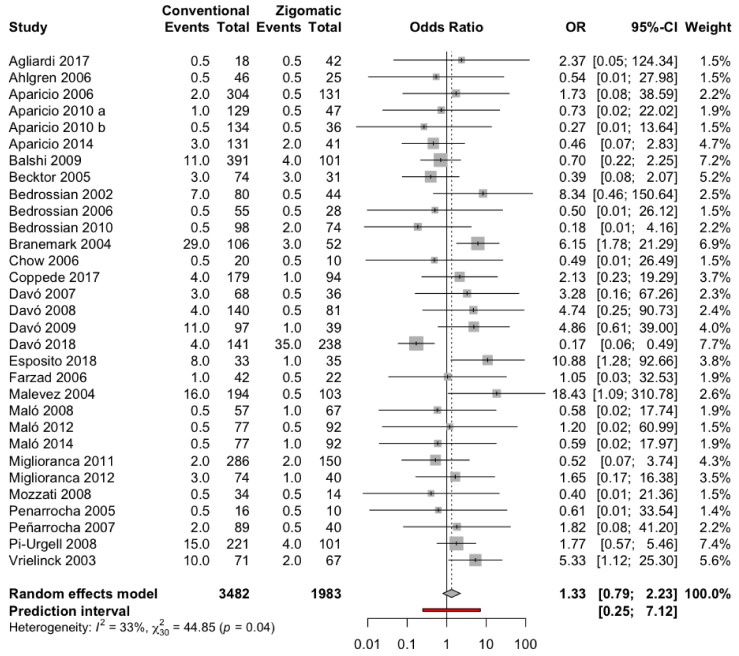
Forest plot of the OR meta-analysis of implant failure: conventional dental implant group versus zygomatic dental implant group.

**Figure 3 biology-10-00601-f003:**
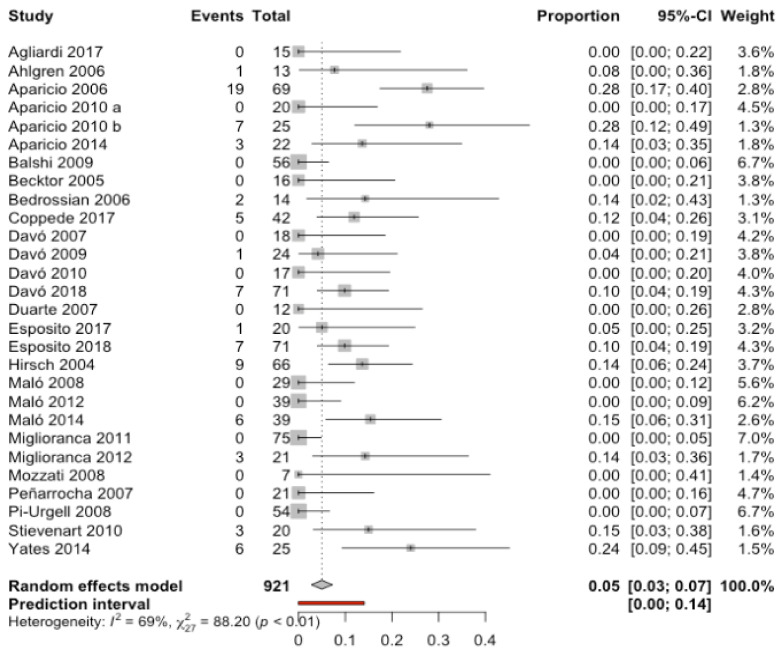
Forest plot of the incidence of prosthetic complications in patients with zygomatic implants.

**Figure 4 biology-10-00601-f004:**
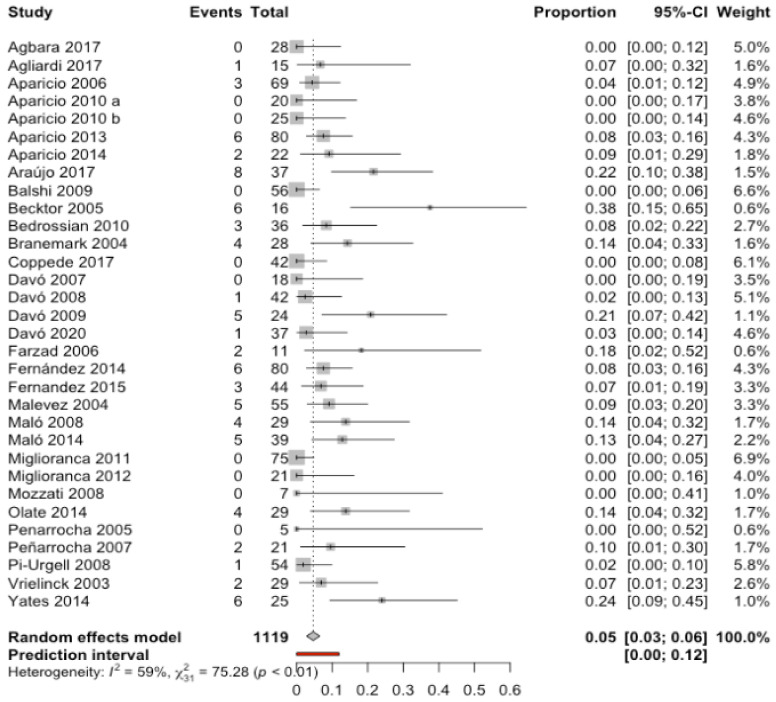
Forest plot of the cumulative incidence of sinus complications in patients with zygomatic implants.

**Figure 5 biology-10-00601-f005:**
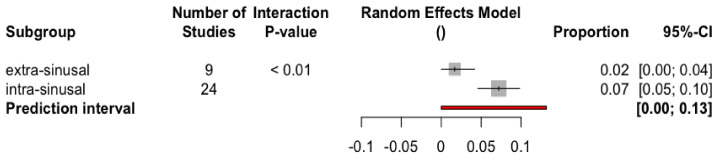
Forest plot of the cumulative incidence of sinus complications in patients with zygomatic implants by subgroup.

**Figure 6 biology-10-00601-f006:**
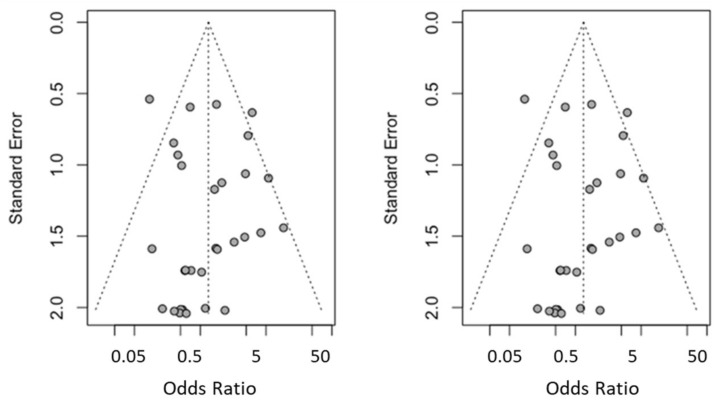
Initial funnel plot and after trim and fill adjustment.

**Table 1 biology-10-00601-t001:** Qualitative analysis of articles included in the systematic review.

Author/Year	Study Type	Sample (*n*)	Follow-up Time (Months)	Implant Failure	Prosthetic Complications	Sinusitis
Agbara et al., 2017 [37]	Retrospective study	42 ZI 123 CI	51.7 (5–163)	5/42 ZI (peri-implantitis (*n* = 4) and accidental intrasinusal placement (*n* = 1))	N/A/28 patients	0/28 patients
				N/A/123 CI		
Agliardi et al., 2017 [21]	NRCT	42 ZI 18 CI	85.04 (73–91)	0/42 ZI 0/18 CI	0/15 patients	1/15 patients: intrasinusal placement of the zygomatic dental implants (sinus membrane perforation close to the bone crest, treated with antiseptics (chlorhexidine 0.2%), antibiotics (amoxicillin and clavulanic acid, 1000 mg) and corticosteroids)
Ahlgren et al., 2006 [22]	NRCT	25 ZI 46 CI	11–49	0/25 ZI 0/46 CI	1/13 patients (allergy to the gold alloy of the overdenture bar)	N/A/13 patients
Aparicio et al., 2006 [24]	NRCT	131 ZI 304 CI	25.1 (6–60)	0/131 ZI	19/69 patients (loosening of the zygomatic implant gold screws (*n* = 9). Fracture of one gold screw (*n* = 2). Fracture of the metal resin prosthesis (*n* = 8))	3/69 patients: intrasinusal placement of the zygomatic dental implants (after 14, 23, and 27 months postsurgery and treated by antibiotics)
				2/304 CI (pterygoid implant failed 1 month after abutment connection (*n* = 1) and an anterior implant failed after 27 months in function (*n* = 1))		
Aparicio et al., 2010a [26]	NRCT	47 ZI 129 CI	24–60	0/47 ZI	7/25 patients (fracture of the teeth of metal–resin (*n* = 4) and metal–porcelain prostheses (*n* = 1) and a fracture of an abutment screw (*n* = 1))	0/25 patients: intrasinusal (*n* = 7) and extrasinusal (*n* = 18) placement of the zygomatic dental implants
				1/129 CI (pterygoid implant failed 52 months of loading (*n* = 1))		
Aparicio et al., 2010b [23]	NRCT	36 ZI 104 CI	36–48	0/36 ZI 0/104 CI	0/20 patients	0/20 patients: extrasinusal placement of the zygomatic dental implants
Aparicio et al., 2013 [35]	NRCT	41 classic procedure ZI 156 ZAGA	135,24 classic procedure ZI 55.44 ZAGA	3/41 classic procedure ZI 0/156 ZAGA	2/197	1/22 patients: intrasinusal technique 1/80 ZAGA technique
Aparicio et al., 2014 [1]	NRCT	41 ZI 131 CI	120	2/41 ZI (extreme peri-implant infection with complete dissolution of the palatal bone)	23/22 patients (fracture of the framework (*n* = 1), loosening of gold ZI screws (*n* = 4), fracture of gold screws (*n* = 4), loosening of the abutment screw (*n* = 3), fracture of ceramic prosthetic teeth (*n* = 5), fracture of resin prostheses (*n* = 2), disconnected abutments (*n* = 4))	5/22 patients: intrasinusal placement of the zygomatic dental implants.Sinusitis was treated with antibiotics
				3/131 CI (anterior implant failure 1 month after abutment connection (*n* = 1) and 3 years of function placed in the subnasal area (*n* = 1), and in the pterygoid area previous to prosthesis installation (*n* = 1))		
Araújo et al., 2014 [52]	Retrospective study	129 ZI	12	2/129 (failures occurred 5–7 months postoperatively)	N/A/37 patients	8/37 patients: sinus slot technique of the zygomatic dental implants
Balshi et al., 2009 [38]	Retrospective study	101 ZI 391 CI	9–60	4/101 ZI 11/391 CI	0/56 patients	N/A/56 patients
Becktor et al., 2005 [7]	Retrospective study	31 ZI 74 CI	46.4 (9–69)	3/31 ZI 3/74 CI	0/16 patients	6/16 patients: intrasinusal placement of the zygomatic dental implants
Bedrossian et al., 2002 [28]	NRCT	44 ZI 80 CI	34	0/44 ZI 7/80 CI	N/A/22 patients	N/A/22 patients
Bedrossian et al., 2006 [8]	Retrospective study	28 ZI 55 CI	12	0/28 ZI 0/55 CI	2/14 patients (partial fractures in the denture around the zygomatic implant cylinder)	N/A/14 patients
Bedrossian et al., 2010 [27]	NRCT	74 ZI 98 CI	84	2/74 ZI 0/98 CI	N/A/36 patients	3/36 patients: intrasinusal placement of the zygomatic dental implants
Boyes-Varley et al., 2003 [30]	NRCT	77 ZI	30	0/77 ZI	N/A/45 patients	N/A/45 patients
Branemark et al., 2004 [29]	NRCT	52 ZI 106 CI	60–120	3/52 ZI 29/106 CI	N/A/28 patients	4/28 patients: intrasinusal placement of the zygomatic dental implants
Chow et al., 2006 [53]	Case series	10 ZI 20 CI	6–10	0/10 ZI 0/20 CI	N/A/5 patients	N/A/5 patients
Coppede et al., 2017 [36]	NRCT	94 ZI 179 CI	36	1/94 ZI 4/179 CI	5/42 patients (five fractures or detachments of one or more acrylic teeth)	0/42 patients: extrasinusal placement of the zygomatic dental implants
Davó et al., 2007 [41]	Retrospective study	36 ZI 68 CI	6–29	0/36 ZI 3/68 CI	0/18 patients	0/18 patients: intrasinusal placement of the zygomatic dental implants
Davó et al., 2008 [9]	Retrospective study	81 ZI 140 CI	12–24	0/81 ZI 4/140 CI	N/A/42 patients	1/42 patients: sinus slot technique (*n* = 15 ZI) and intrasinusal placement of the zygomatic dental implants (*n* = 66 ZI)
Davó, 2009 [42]	Retrospective study	39 ZI 97 CI	60	1/39 ZI 11/97 CI	1/24 patients	5/24 patients C
Davó et al., 2010 [14]	NRCT	68 ZI	12	0/68 ZI	0/17 patients	N/A/17 patients
Davó et al., 2018 [17]	RCT	238 ZI 141 CI	6	35/238 ZI 4/141 CI	7/71 patients	N/A/71 patients
Davó et al., 2020 [39]	Retrospective study	182 ZI	10.5	0/182 ZI	N/A/37 patients	1/37 patients: intrasinusal (6%) and extrasinusal placement of the zygomatic dental implants (94%)
Duarte et al.,2007 [10]	NRCT	48 ZI	6–30	2/48 ZI	0/12 patients	N/A/12 patients
Esposito et al., 2017 [18]	RCT	80 ZI	12	2/80 ZI	1/20 patients (fracture of provisional prosthesis)	N/A/20 patients
Esposito et al., 2018 [19]	RCT	35 ZI 33 CIAug	4	1/35 ZI 8/33 CIAug	7/71 patients	N/A/71 patients
Farzad et al., 2006 [32]	NRCT	22 ZI 42 CI	18–46	0/22 ZI 1/42 CI	N/A/11 patients	2/11 patients: intrasinusal placement of the zygomatic dental implants
Fernández et al., 2014 [47]	Retrospective study	244 ZI	6–48	1/244 ZI	N/A/80 patients	6/80 patients: intrasinusal placement of the zygomatic dental implants
Fernández et al., 2015 [20]	RCT	41 ZI	3	1/19 ZI without inferior meatal antrostomy 0/22 ZI with inferior meatal antrostomy	N/A/44 patients	3/44 patients: intrasinusal placement of the zygomatic dental implants without inferior meatal antrostomy
Hirsch et al., 2004 [31]	NRCT	124 ZI	12	3/124 ZI	9/66 patients	N/A/66 patients
Malevez et al., 2004 [13]	Retrospective study	103 ZI 194 CI	6–48	0/103 ZI 16/194 CI	N/A/55 patients	5/55 patients: intrasinusal placement of the zygomatic dental implants
Maló et al., 2008 [11]	NRCT	67 ZI 57 CI	13(6–18)	1/67 ZI 0/57 CI	0/29 patients	4/29 patients: extrasinusal placement of the zygomatic dental implants
Maló et al., 2012 [44]	Retrospective study	92 ZI 77 CI	36	0/92 ZI 0/77 CI	0/39 patients	5/39 patients: extrasinusal placement of the zygomatic dental implants
Maló et al., 2014 [43]	Retrospective study	92 ZI 77 CI	60	1/92 ZI 0/77 CI	6/39 patients	5/39 patients: extrasinusal placement of the zygomatic dental implants
Miglioranca et al., 2011 [46]	Retrospective study	150 ZI 286 CI	12	2/150 ZI 2/286 CI	0/75 patients	0/75 patients: extrasinusal placement of the zygomatic dental implants
Miglioranca et al., 2012 [34]	NRCT	40 ZI 74 CI	96	1/40 ZI 3/74 CI	3/21 patients (the metal bar was broken in patient 8; 2 patients reported difficulty in cleaning around the abutment connected to the zygomatic implant)	0/21 patients: extrasinusal placement of the zygomatic dental implants
Mozzati et al., 2008 [54]	Case series	14 ZI 34 CI	24	0/14 ZI 0/34 CI	0/7 patients	0/7 patients: intrasinusal placement of the zygomatic dental implants
Rodríguez-Chessa, 2014 [45]	Retrospective study	67 ZI 84 CI	20	14/67 ZI N/A/84 CI	N/A/29 patients	4/29 patients: sinus slot technique of the zygomatic dental implants
Peñarrocha et al., 2005 [55]	Case series	10 ZI 16 CI	12–18	0/10 ZI 0/16 CI	N/A/5 patients	0/5 patients: sinus slot technique of the zygomatic dental implants
Peñarrocha et al., 2007 [48]	Retrospective study	40 ZI 89 CI	29 (12–45)	0/40 ZI 2/89 CI	0/21 patients	2/21 patients: sinus slot technique of the zygomatic dental implants
Peñarrocha-Diago et al., 2020 [51]	Retrospective study	31 ZI	12	0/31 ZI	N/A/19 patients	N/A/19 patients
Pi-Urgell et al., 2008 [42]	Retrospective study	101 ZI 221 CI	1–72	4/101 ZI 15/221 CI	0/54 patients	1/54 patients: intrasinusal placement of the zygomatic dental implants
Stievenart et al., 2010 [49]	Retrospective study	80 ZI	6–40	3/80 ZI	3/20 patients	N/A/20 patients
Vrielinck et al., 2003 [33]	NRCT	67 ZI 71 CI	24	2/67 ZI 10/71 CI	N/A/29 patients	2/29 patients: intrasinusal placement of the zygomatic dental implants
Yates et al., 2014 [50]	Retrospective study	43 ZI	60–120	6/43 ZI	6/25 patients	6/25 patients: sinus slot technique of the zygomatic dental implants

NRCT: Nonrandomized Clinical Trial; RCT: Randomized Controlled Trial; CT: Controlled Trial; CS: Case Series; N/A: Not Available; ZI: Zygomatic Implants; CI: Conventional Implants; CIAug: Conventional Implants with Bone Augmentation; ZAGA: Zygomatic Anatomy-Guided Approach.

**Table 2 biology-10-00601-t002:** Assessment of methodological quality according to the Jadad scale.

Jadad Criteria						
Author/Year	Is the Study Described as Randomized?	Is the Study Described as Double-Blinded?	Was There a Description of Withdrawals and Dropouts?	Was the Method of Randomization Adequate?	Was the Method of Blinding Appropriate?	Score
Agbara et al., 2017 [37]	N/A	N/A	N/A	N/A	N/A	N/A
Agliardi et al., 2017 [21]	0	0	0	0	0	0
Ahlgren et al., 2006 [22]	0	0	0	0	0	0
Aparicio et al., 2006 [24]	0	0	0	0	0	0
Aparicio et al., 2010a [26]	0	0	0	0	0	0
Aparicio et al., 2010b [23]	0	0	0	0	0	0
Aparicio et al., 2013 [35]	0	0	0	0	0	0
Aparicio et al., 2014 [1]	0	0	0	0	0	0
Araújo et al., 2017 [52]	N/A	N/A	N/A	N/A	N/A	N/A
Balshi et al., 2009 [38]	N/A	N/A	N/A	N/A	N/A	N/A
Becktor et al., 2005 [7]	N/A	N/A	N/A	N/A	N/A	N/A
Bedrossian et al., 2002 [28]	0	0	0	0	0	0
Bedrossian et al., 2006 [8]	N/A	N/A	N/A	N/A	N/A	N/A
Bedrossian et al., 2010 [27]	0	0	0	0	0	0
Boyes-Varley et al., 2003 [30]	0	0	0	0	0	0
Branemark et al., 2004 [29]	0	0	0	0	0	0
Chow et al., 2006 [53]	N/A	N/A	N/A	N/A	N/A	N/A
Coppede et al., 2017 [36]	0	0	0	0	0	0
Davó et al., 2007 [41]	N/A	N/A	N/A	N/A	N/A	N/A
Davó et al., 2008 [9]	N/A	N/A	N/A	N/A	N/A	N/A
Davó, 2009 [42]	N/A	N/A	N/A	N/A	N/A	N/A
Davó et al., 2010 [14]	0	0	1	0	0	1
Davó et al., 2018 [17]	1	1	1	1	1	5
Davó et al., 2020 [39]	N/A	N/A	N/A	N/A	N/A	N/A
Duarte et al.,2007 [10]	0	0	0	0	0	0
Esposito et al., 2017 [18]	1	0	1	1	0	3
Esposito et al., 2018 [19]	1	0	1	1	0	3
Farzad et al., 2006 [32]	0	0	0	0	0	0
Fernández et al., 2014 [47]	N/A	N/A	N/A	N/A	N/A	N/A
Fernández et al., 2015 [20]	1	1	1	1	1	5
Hirsch et al., 2004 [31]	0	0	0	0	0	0
Malevez et al., 2004 [13]	N/A	N/A	N/A	N/A	N/A	N/A
Maló et al., 2008 [11]	0	0	0	0	0	0
Maló et al., 2012 [44]	N/A	N/A	N/A	N/A	N/A	N/A
Maló et al., 2014 [43]	N/A	N/A	N/A	N/A	N/A	N/A
Miglioranca et al., 2011 [46]	N/A	N/A	N/A	N/A	N/A	N/A
Miglioranca et al., 2012 [34]	0	0	1	0	0	1
Mozzati et al., 2008 [54]	N/A	N/A	N/A	N/A	N/A	N/A
Rodríguez-Chessa et al., 2014 [45]	N/A	N/A	N/A	N/A	N/A	N/A
Peñarrocha et al., 2005 [55]	N/A	N/A	N/A	N/A	N/A	N/A
Peñarrocha et al., 2007 [48]	N/A	N/A	N/A	N/A	N/A	N/A
Peñarrocha-Diago et al., 2020 [51]	N/A	N/A	N/A	N/A	N/A	N/A
Pi-Urgell et al., 2008 [42]	N/A	N/A	N/A	N/A	N/A	N/A
Stievenart et al., 2010 [49]	N/A	N/A	N/A	N/A	N/A	N/A
Vrielinck et al., 2003 [33]	0	0	0	0	0	0
Yates et al., 2014 [50]	N/A	N/A	N/A	N/A	N/A	N/A

N/A: Not applicable.

## Data Availability

Data are available on request due to restrictions, e.g., privacy or ethical.

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
