# Peer review of "Survival Rate and Prosthetic and Sinus Complications of Zygomatic Dental Implants for the Rehabilitation of the Atrophic Edentulous Maxilla: A Systematic Review and Meta-Analysis"

_biology, 2021, doi:10.3390/biology10070601_

Round 1

Reviewer 1 Report

Dear Authors,

This study (Systematic Review and Meta-Analysis) described literature search results of the survival rate, prosthetic and sinus complications of zygomatic dental implants for the rehabilitation of the atrophic edentulous maxilla.

It concluded that zygomatic implants are recommended for the use in atrophic edentulous maxilla rehabilitation due to the high survival rate and low prosthetic and sinus complications compared to conventional dental implants.

I believe that the manuscript with several minor corrections will be improved, and this paper will be useful in developing guideline for implant treatment in future.

I have several comments for the authors.

  1. Authors should be precisely describing the number of zygomatic and conventional dental implants used in the rehabilitation of the atrophic edentulous maxilla from literature.
  2. In discussion (page15 line271-), authors mentioned that laser therapy including photodynamic therapy. It is irrelevant and inappropriate with the data collected.  Also you should check the reference again.
  3. In discussion, authors mentioned that the length of dental implants related survival rate. Please add the discussion paragraph describing the difference in the average lengths of zygomatic dental implants compared with that of conventional dental implants
  4.  Please make sure there were no typo and grammar errors.                   For example: page 2 line 86 `thro ugh` should be `through`.

Author Response

Dear Reviewer 1:

I’m pleased to resubmit the manuscript of the work entitled, “Survival Rate, Prosthetic and Sinus Complications of Zygomatic Dental Implants for the Rehabilitation of the Atrophic Edentulous Maxilla. A Systematic Review and Meta-Analysis “

Reviewer 1: English language and style are fine/minor spell check required

Response: In order to adapt to the reviewer's 1 comments, we have send the manuscript to the English Editing Service of MDPI. We attached the Certificate.

Reviewer 1: Authors should be precisely describing the number of zygomatic and conventional dental implants used in the rehabilitation of the atrophic edentulous maxilla from literature.

Response: In order to adapt to the reviewer's 1 comments, we have added a sentence in the Discussion section, describing the number of zygomatic and conventional dental implants used in the rehabilitation of the atrophic edentulous maxilla

Reviewer 1: In discussion (page15 line271-), authors mentioned that laser therapy including photodynamic therapy. It is irrelevant and inappropriate with the data collected.  Also you should check the reference again.

Response: In order to adapt to the reviewer's 1 comments, we have removed the sentence.

Reviewer 1: In discussion, authors mentioned that the length of dental implants related survival rate. Please add the discussion paragraph describing the difference in the average lengths of zygomatic dental implants compared with that of conventional dental implants

Response: In order to adapt to the reviewer's 1 comments, we have added a paragraph describing the difference in the average lengths of zygomatic dental implants compared with that of conventional dental implants.

Reviewer 1: Please make sure there were no typo and grammar errors.                   For example: page 2 line 86 `thro ugh` should be `through`.

Response: In order to adapt to the reviewer's 1 comments, we have corrected the grammar errors and also send the manuscript to the English Editing Service of MDPI. We attached the Certificate.

We take this opportunity to thank the recommendations and suggestions made by the reviewer to improve the document.

Yours sincerely,

Reviewer 2 Report

Although the PICO question in this study involved comparing zygomatic and conventional dental implants in terms of the following outcomes survival rate, prosthetic and sinus complications, the authors only compared the two types of implant placement for survival rate in their meta analysis. Can the authors provide an explanation on why the comparison was not performed for prosthetic and sinus complications too?

Author Response

Dear Reviewer 2:

I’m pleased to resubmit the manuscript of the work entitled, “Survival Rate, Prosthetic and Sinus Complications of Zygomatic Dental Implants for the Rehabilitation of the Atrophic Edentulous Maxilla. A Systematic Review and Meta-Analysis “

Reviewer 2: English language and style are fine/minor spell check required

Response: In order to adapt to the reviewer's 2 comments, we have send the manuscript to the English Editing Service of MDPI. We attached the Certificate.

Reviewer 2: The authors only compared the two types of implant placement for survival rate in their meta analysis. Can the authors provide an explanation on why the comparison was not performed for prosthetic and sinus complications too?

Response: In order to adapt to the reviewer's 2 comments, we clarify that it is possible to provide information about the survival rate of two dental implant lengths; however, the prosthetic rehabilitation includes both types of implant and it is not possible to associate complications to one of them. In addition, the conventional length dental implants are far away from the maxillary sinus (premaxilla), so it is not associate to maxillary sinus damage.

We take this opportunity to thank the recommendations and suggestions made by the reviewer to improve the document.

Yours sincerely,

Reviewer 3 Report

In the manuscript entitled: “Survival Rate, Prosthetic and Sinus Complications of Zygomatic Dental Implants for the Rehabilitation of the Atrophic Edentulous Maxilla. A Systematic Review and Meta-Analysis”, the authors identified the survival rate, prosthetic and sinus complications of zygomatic dental implants for the rehabilitation of the atrophic edentulous maxilla.

The authors found that Conventional dental implants failure (n = 3549) has been established in 2.89%, while zygomatic dental implants failure (n = 1895) has been established in 0.69%. The measure of the effect size used has been the 29 Odds Ratio, which has been estimated at 2.05 with a confidence interval of 95% between 1.22 and 3.44. The failure risk of conventional dental implant has stated 2.1 times higher than zygomatic dental implants.

The authors concluded that the high survival rate and low prosthetic and sinus complications related to 38 zygomatic dental implants recommend the use of zygomatic dental implants for the rehabilitation 39 of the atrophic edentulous maxilla.

Major comments:

In general, the idea and innovation of this study, regards analysis of Dental Implants for the Rehabilitation of the Atrophic Edentulous Maxilla is interesting, because the role these factors in dentistry are validated but further studies on this topic could be an innovative issue in this field could be open a creative matter of debate in literature by adding new information. Moreover, there are few reports in the literature that studied this interesting topic with this kind of study design.

The study was well conducted by the authors; However, there are some concerns to revise that are described below.

The introduction section resumes the existing knowledge regarding the important factor linked with implant failure.

However, as the importance of the topic, the reviewer strongly recommends, before a further re-evaluation of the manuscript, to update the literature through read, discuss and must cites in the references with great attention all of those recent interesting articles, that helps the authors to better introduce and discuss the role of periodontitis and related biomarkers as cause of implant failure (Galectin, NLRP3): 1) Isola G, Polizzi A, Alibrandi A, Williams RC, Lo Giudice A. Analysis of galectin-3 levels as a source of coronary heart disease risk during periodontitis. J Periodontal Res. 2021 Feb 28. doi: 10.1111/jre.12860. 2) Isola G, Polizzi A, Santonocito S, Alibrandi A, Williams RC. Periodontitis activates the NLRP3 inflammasome in serum and saliva. J Periodontol. 2021 May 19. doi: 10.1002/JPER.21-0049.

The authors should be better specified, at the end of the introduction section, the rational of the study and the aim of the study. In the material and methods section, should better clarify the risk of biase. Moreover, please more specifiy the clinicians involved in the different stages of the review.

The discussion section appears well organized with the relevant paper that support the conclusions, even if the authors should better discuss the relationship between periodontitis and implant failure. The conclusion should reinforce in light of the discussions.

In conclusion, I am sure that the authors are fine clinicians who achieve very nice results with their adopted protocol. However, this study, in my view does not in its current form satisfy a very high scientific requirement for publication in this journal and requests a revision before a futher re-evaluation of the manuscript.

Minor Comments:

Abstract:

  • Better formulate the abstract section by better describing the aim of the study

Introduction:

  • Please refer to major comments

Discussion

  • Please add a specific sentence that clarifies the results obtained in the first part of the discussion
  • Page 15 last paragraph: Please reorganize this paragraph that is not clear

Author Response

Dear Reviewer 3:

I’m pleased to resubmit the manuscript of the work entitled, “Survival Rate, Prosthetic and Sinus Complications of Zygomatic Dental Implants for the Rehabilitation of the Atrophic Edentulous Maxilla. A Systematic Review and Meta-Analysis “

Reviewer 3: English language and style are fine/minor spell check required

Response: In order to adapt to the reviewer's 3 comments, we have send the manuscript to the English Editing Service of MDPI. We attached the Certificate.

Reviewer 3: However, as the importance of the topic, the reviewer strongly recommends, before a further re-evaluation of the manuscript, to update the literature through read, discuss and must cites in the references with great attention all of those recent interesting articles, that helps the authors to better introduce and discuss the role of periodontitis and related biomarkers as cause of implant failure (Galectin, NLRP3): 1) Isola G, Polizzi A, Alibrandi A, Williams RC, Lo Giudice A. Analysis of galectin-3 levels as a source of coronary heart disease risk during periodontitis. J Periodontal Res. 2021 Feb 28. doi: 10.1111/jre.12860. 2) Isola G, Polizzi A, Santonocito S, Alibrandi A, Williams RC. Periodontitis activates the NLRP3 inflammasome in serum and saliva. J Periodontol. 2021 May 19. doi: 10.1002/JPER.21-0049.

Response: In order to adapt to the reviewer's 3 comments, we have added the sentence introducing and discussing the role of periodontitis and related biomarkers as cause of implant failure.

Reviewer 3: The authors should be better specified, at the end of the introduction section, the rational of the study and the aim of the study.

Response: In order to adapt to the reviewer's 3 comments, we have better specified the rational and the aim of the study

Reviewer 3: In the material and methods section, should better clarify the risk of biase

Response: In order to adapt to the reviewer's 3 comments, we clarify that the risk of bias has been analyzed by JADAD scale.

Reviewer 3: Moreover, please more specifiy the clinicians involved in the different stages of the review.

Response: In order to adapt to the reviewer's 3 comments, we have specified the clinicians involved in the different stages of the review.

Reviewer 3: The discussion section appears well organized with the relevant paper that support the conclusions, even if the authors should better discuss the relationship between periodontitis and implant failure.

Response: In order to adapt to the reviewer's 3 comments, we have added a sentence to discuss the relationship between periodontitis and implant failure.

Reviewer 3: The conclusion should reinforce in light of the discussions.

Response: In order to adapt to the reviewer's 3 comments, we appreciate the reviewer's comment, but we consider that the conclusion should respond to the objectives of the study.

Reviewer 3: Abstract: Better formulate the abstract section by better describing the aim of the study

Response: In order to adapt to the reviewer's 3 comments, we have tried to better describe the aim of the study in the Abstract section.

Reviewer 3: Discussion: Please add a specific sentence that clarifies the results obtained in the first part of the discussion

Response: In order to adapt to the reviewer's 3 comments, we have added a specific sentence that clarifies the results obtained in the first part of the discussion

Reviewer 3: Discussion: Page 15 last paragraph: Please reorganize this paragraph that is not clear

Response: In order to adapt to the reviewer's 3 comments, we have reorganized the paragraph.

We take this opportunity to thank the recommendations and suggestions made by the reviewer to improve the document.

Yours sincerely,

Round 2

Reviewer 3 Report

The authors have well addressed to all reviewer's comments. The manuscript can be accepted